# Risk Factors for Exposure of Wild Birds to West Nile Virus in A Gradient of Wildlife-Livestock Interaction

**DOI:** 10.3390/pathogens12010083

**Published:** 2023-01-03

**Authors:** Laia Casades-Martí, Rocío Holgado-Martín, Pilar Aguilera-Sepúlveda, Francisco Llorente, Elisa Pérez-Ramírez, Miguel Ángel Jiménez-Clavero, Francisco Ruiz-Fons

**Affiliations:** 1Health & Biotechnology (SaBio) Group, Instituto de Investigación en Recursos Cinegéticos (IREC), CSIC-UCLM-JCCM, 13005 Ciudad Real, Spain; 2Centro de Investigación en Sanidad Animal (CISA), INIA-CSIC, 28130 Valdeolmos, Spain; 3Centro de Investigación Biomédica en Red de Epidemiología y Salud Pública (CIBERESP), Instituto de Salud Carlos III, 28029 Madrid, Spain; 4CIBERINFEC—CIBER de Enfermedades Infecciosas, Centro de Investigación Biomédica en Red de Enfermedades Infecciosas, Instituto de Salud Carlos III, 28029 Madrid, Spain

**Keywords:** bird diversity, disease ecology, emerging zoonoses, *Flavivirus*, horse, risk factors

## Abstract

West Nile virus (WNV) transmission rate is shaped by the interaction between virus reservoirs and vectors, which may be maximized in farm environments. Based on this hypothesis, we screened for WNV in wild birds in three scenarios with decreasing gradient of interaction with horses: (i) the farm (A_1_); (ii) the neighborhood (A_2_); and (iii) a wild area (A_3_). We captured wild birds and analyzed their sera for WNV antibodies by blocking ELISA and micro-virus neutralization test. *Flavivirus* infections were tested with generic and specific PCR protocols. We parameterized linear mixed models with predictors (bird abundance and diversity, vector abundance, vector host abundance, and weather quantities) to identify *Flavivirus* spp. and WNV exposure risk factors. We detected a low rate of *Flavivirus* infections by PCR (0.8%) and 6.9% of the birds were seropositive by ELISA. Exposure to *Flavivirus* spp. was higher in A_1_ (9%) than in A_2_ and A_3_ (5.6% and 5.8%, respectively). Bird diversity was the most relevant predictor of exposure risk and passerines dominated the on-farm bird community. Our results suggest that measures deterring the use of the farm by passerines should be implemented because the environmental favorability of continental Mediterranean environments for WNV is increasing and more outbreaks are expected.

## 1. Introduction

Flaviviruses are emerging and re-emerging arboviruses of public and animal health relevance. They have a high dispersal capacity which allows them to potentially expand their original spatial range and emerge in new areas, causing disease outbreaks with high impact on wildlife, livestock, and human health [1]. The spatial distribution and the number of reported infections of mosquito-borne flaviviruses (e.g., West Nile virus, WNV, Usutu virus, USUV, and Japanese encephalitis virus, JEV) have remarkably increased in recent decades worldwide [2]. Many of the known flaviviruses are pathogenic for animals, and over 50% of them are also pathogenic for humans, e.g., JEV, WNV, yellow fever virus, dengue virus, Zika virus, or tick-borne encephalitis virus, among others. Clinical signs caused by *Flavivirus* spp. infections usually include neurological lesions of encephalitis or meningoencephalitis [3]. Most flaviviruses present limited spatial ranges and circulate enzootically in a specific host-vector network of interactions [4]. However, particular flaviviruses, e.g., WNV, have been able to expand worldwide and have established enzootic cycles in contrasting communities of hosts and vectors, thereby currently being of worldwide concern. Among the most clinically important emerging flaviviruses in Europe are those belonging to the JEV sero-complex group, including WNV and USUV [5]. Other mosquito-borne flaviviruses have also been detected in Europe, e.g., JEV [6], Bagaza virus (BAGV) [7], and specific endogenous flaviviruses of mosquitoes [8]. The viruses of the JEV group present common ecological traits, such as (1) being mainly avian flaviviruses (birds are their natural reservoirs), (2) being transmitted by mosquitoes, with *Culex* spp. as most competent vectors, (3) being able of infecting other vertebrate hosts, including several mammalian species, and (4) potentially causing severe viral encephalitis outbreaks with high case fatality [9].

Migratory birds play an essential role in the long-distance movement of JEV sero-complex flaviviruses and are deemed as the pathway that favored the introduction of WNV into Europe. When migrations occur between enzootic and virus-free areas, birds that become infected prior to or during migration can actively carry the virus in their blood (and other tissues) and infect mosquitoes and/or their predators [10] in destination territories. Introductions into ecological settings with favorable conditions for local maintenance, i.e., presence and abundance of competent birds and mosquitoes and favorable abiotic conditions, are key factors for their establishment in new territories [11]. Resident birds in enzootic territories also play a very important role in the dispersal of flaviviruses at smaller spatial scales but greater than the distances that infected mosquitoes can reach [12]. Particular resident bird species have higher sensitivity to different flaviviruses, e.g., waterfowl, corvids, or birds of prey are sensitive to WNV [13]. However, several other species of birds show no clinical evidence of infection and participate in the local maintenance and spread of these flaviviruses with competent mosquitoes [14]. *Culex* spp. mosquitoes are major vectors of WNV and USUV. *Culex* spp. breed in stagnant, calm, and shallow waters such as ponds, but some species, e.g., *Cx. pipiens*, breed in different sources of stagnant water in very different environments [15]. The presence of birds replicating JEV group flaviviruses is important for enzootic circulation, but the presence and abundance of mosquitoes feeding on birds and being able to replicate and transmit the viruses is paramount [11]. 

West Nile virus was reported as the cause of death of a resident raptor in south-central Spain in 2007 [16], but it was not until 2010 that Spain notified the first cases of West Nile fever (WNF) in horses and humans [17]. Since then, WNF cases have been reported in wild birds, horses, and humans [1], indicating that WNV circulates enzootically in wide areas in Spain. The virus may probably present a wider extension in the country and its presence and impact could remain underestimated [18]. Since 2006, mosquitoes [19] and wild birds [20] have been found carrying USUV RNA. Further, BAGV was first detected in southern Spain in 2010 [7]. Competent WNV, USUV, and BAGV birds and mosquitoes coexist in large territories of peninsular Spain, creating the ideal breeding ground for the emergence of these viruses. The recent emergence of WNF in western Spain [17], where previous evidence showed enzootic circulation [16], suggests that favorable conditions for virus circulation have increased in these areas, also in the vicinity of equine farms. Understanding the risks for horses (and people in or close to the farms) is essential to prevent (e.g., by targeted vaccination) and control (e.g., by reducing vector mosquito populations or deterring attraction to wild birds) WNF cases, especially if, under these assumptions, and with increasing winter temperatures in inland Spain, the prevalence of WNV circulation increases in the future. In this scenario, the aim of this study was to estimate the drivers of WNV circulation in a gradient of interaction between birds, mosquitoes, and horses that result in different host community assemblages and in potential variations in the community of vectors. We evaluated WNV circulation rates in the studied scenarios by estimating the prevalence of WNV antibodies in wild birds, the drivers of exposure as indicators of WNV circulation rates, and the risk of transmission to horses and humans in inner continental Spain.

## 2. Materials and Methods

### 2.1. Sampling Area

We selected nine equine farms in south-central continental Spain where WNF cases had been notified in 2007, 2014 and 2015 [17]. Study farms were also selected to be representative of the continental Mediterranean ecosystems in which WNV infection has been reported and thus account for potential variations in bird and mosquito communities having local effects on the circulation of WNV. Previous findings demonstrated that the mosquito communities in continental areas of south-central Spain are dominated by *Cx. pipiens* and *Cx. theileri* [21], thus showing that WNV vectors are present in the region. Four of the selected locations were close to areas where WNF cases were notified in horses in 2014 in southern Ciudad Real province (Figure 1), Castilla-La Mancha (CLM). Four additional locations were selected in the north-west of Toledo province (CLM) close to the WNF outbreak reported in pheasants in 2015. An additional horse farm was selected in the south-center of Toledo province; WNV was isolated from clinically infected golden eagles (*Aquila chrysaetos*) in similar landscapes of southern Toledo province [16]. The predominant climate in these areas is continental Mediterranean, characterized by cold winters and very hot summers, and with an average annual rainfall of 342 mm in Toledo province (Spanish Meteorological Agency (AEMET); Period: 1982–2010. Altitude: 515 m. Latitude: 39°53′5″ N, Longitude: 4°2′43″ W) and 402 mm in Ciudad Real province (AEMET; Period: 1981–2010. Altitude: 628 m. Latitude: 38°59′21″ N, Longitude: 3°55′13″ W). Mediterranean woodlands of holm oak (*Quercus ilex*), cork oak (*Q. suber*) and gall oak (*Q. faginea*) predominate in these environments, interspersed with extensive patches of grassland and Mediterranean shrubland (*Cystus* spp., *Rosmarinus* spp., *Erica* spp. and *Philyrea* spp.). 

On each of the nine farms, three areas were selected to account for a variable gradient of wildlife-livestock interaction that could drive the risk of WNV infection in the farm environment (Figure 2), either from wild birds on the farm or from birds in the vicinity of the farm. These areas included: (i) the farm environment where horses interact with wild birds (Area 1, A_1_); (ii) an area in-between the farm and a wildlife-dominated area at a 0.5-1 km linear distance to the farm where interactions are feasible but limited (Area 2, A_2_); and iii) a wildlife-dominated area at a 3–5 km linear distance from the farm with no wildlife-livestock interactions (Area 3, A_3_). Distance ranges separating the study areas were selected based on known daily home ranges of *Culex* spp. mosquitoes that rarely fly few hundred meters away from birth places, and the maximum reported travelling distance of 5 km [12]. This approach was undertaken with the dual intention of (1) analyzing the influence of the local WNV mosquito vector population, and (2) being able to infer risks to equine farms from the neighborhood at distances easily bridged by resident birds and, eventually, by mosquito vectors.

### 2.2. Sample Collection

For logistical reasons, bird surveys could only be performed in five of the selected equine farms and their environment: (i) two (S_1_ and S_2_) in southern Ciudad Real province; (ii) one (S_3_) in the south of Toledo province; and iii) two (S_4_ and S_5_) in north-western Toledo province (Figure 1). Between July 2018 and October 2019, wild birds were captured in survey areas (N = 15) using mist nets (ECOTONE 1016/12, Oryx, Barcelona, Spain). Capture trials were conducted in two different periods per location and area. To maximize bird captures, we selected a water pond or stream per survey area to which local birds are attracted and allocated eighteen linear meters of mist nets to one of its sides. We set the nets before sunset on the first sampling day and kept them active from one hour before sunset until one hour after sunset. The nets remained folded overnight and were deployed again one hour before sunrise on the second day, remaining active until 2 h after sunrise. 

Data (date, point, species and body weight—estimated to the gram tenth with a precision balance) were gathered from the captured birds. We also collected blood samples by puncture of the brachial vein with sterile 23G/25G needles and 0.5 mL/1 mL syringes, depending on bird size, and transferred them into sterile heparinized tubes. Growing feathers, when available, were collected with tweezers into sterile tubes. Oral and cloacal swabs were collected and inserted into tubes containing embedded sponges with virus preservation medium (Deltalab, Barcelona, Spain). All birds were in good condition and responded actively to stimuli, so they were released immediately after sampling. All the samples were kept refrigerated during transport, processed immediately in the lab and stored at −80 °C. Blood was centrifuged for 10′ at 10,000× *g* and the plasma was preserved at −20 °C.

### 2.3. Serological Analyses

Plasma samples were analyzed for the presence of antibodies using a commercial multi-species blocking enzyme-linked immunosorbent assay (bELISA; INGEZIM West Nile COMPAC^®^, Ingenasa, Madrid, Spain) and following the manufacturer’s recommendations. The bELISA detects antibodies against an epitope of the Pr-E protein of WNV and it displays high sensitivity to detect anti-WNV antibodies [22]. Cross-reactivity with other JEV group flaviviruses, e.g., USUV, may occur in this bELISA, but this commercial test displays higher specificity for WNV than other available serological assays [23]. The small volumes of blood that could be obtained from a major part of the birds, without impairing their survival after release, prevented us from specifically assigning the detected antibodies to different JEV group flaviviruses by micro-virus neutralization tests (VNT); only five bELISA positive samples were analyzed by VNT [23] against WNV (NY99, GenBank accession no. KC407666) and USUV (SAAR-1776, GenBank accession no. AY453412.1).

Due to cross-reactions with other flaviviruses in the bELISA, and in order to estimate a true prevalence of WNV exposure in the study areas and WNV determinants, we performed a search in the main science browser engines (Scopus, PubMed and Web of Science) using the terms ‘ELISA’, ‘INGEZIM’, ‘COMPAC’, ‘Flavivirus’, ‘West Nile’, ‘Usutu’ and ‘Bagaza’, in any possible combination among them. After this process, carried out independently by two of the authors, we selected the available articles that met the following criteria: (1) using the commercial bELISA INGEZIM West Nile COMPAC; and (2) testing with a compared VNT a high percentage of the positive sera in the bELISA at least against WNV and USUV, the two most prevalent flaviviruses of the JEV group in Spain. Once the studies were selected, we finally considered seven studies performed on wild birds (six in Spain and one in Poland), one study performed on wild ruminants, another one performed on dogs, another one on horses and another one on zoo mammals, all of them in Spain. We extracted the comparative results of VNT and estimated the proportion of sera confirmed as WNV infection in VNT versus other *Flavivirus* spp. (Appendix A). 

### 2.4. Real-Time RT-PCR Analyses

Total RNA was purified from tissue samples, mainly pulp of growing feather cannons, but also oral and cloacal swabs as previously recommended [13]. Feather cannons were longitudinally dissected in a Class-II cabinet and the pulp was extracted into a nuclease-free sterile tube filled with 350 μL of the extraction buffer of the employed commercial kit (RP1) and 3.5 μL of β-mercaptoethanol (Sigma-Aldrich, Burlington, USA), and vortexing tubes for 1 min. The commercial Macherey-Nagel NucleoSpin TriPrep kit (Fisher Scientific, Düren, Germany) was used for RNA purification following the manufacturer’s protocol. RNA concentration and quality were analyzed using a Nanodrop Spectrophotometer (NanoDropTM One, Thermo Fisher Scientific Inc., Waltham, MA, USA) and five 10 μL aliquots/sample were preserved frozen at −80 °C. 

RNA samples were analyzed using a duplex real-time RT-PCR (rRT-PCR) protocol for the detection of flaviviruses of the JEV and Ntaya (BAGV, Tembusu) sero-complexes [24]. The commercial AgPath-ID One Step RT-PCR kit (Thermo Fisher Scientific Inc., Waltham, MA, USA) was employed for amplifications. Synthetic DNA positive controls (GenScript Biotech Corporation, Rijswijk, The Netherlands) specifically designed for the amplification targets of the 2 primer pairs of the duplex rRT-PCR were used. Purification of viral DNA was performed directly from the PCR product using a purification kit (QIAquick^®^ PCR Purification Kit, Qiagen, Hilden, Germany). Purified samples were sent for Sanger sequencing to Eurofins Genomics GmbH (Ebersberg, Germany). The sequences obtained were compared using the NCBI BLAST tool (Megablast, nucleotide collection database (nr/nt) database). Samples from birds that reacted positive in the duplex rRT-PCR were analyzed by a multiplex real-time RT-PCR (mrRT-PCR) to differentiate between WNV lineages 1 and 2 and USUV [25]. 

A partial 382bp sequence obtained was compared with 35 full genome sequences belonging to different lineages of USUV available in GenBank. Multiple alignments were performed by Clustal W. Phylogenetic trees were generated by the maximum likelihood algorithm available in MEGA7, using K2 + G as the optimal nucleotide substitution model. Bootstrap values were inferred for 1000 replicates.

### 2.5. Predictors of Flavivirus Exposure Risk

#### 2.5.1. Bird Population Diversity and Abundance

The competence of birds in WNV replication and transmission is variable [13]. The orders Passeriformes, Charadriiformes and Strigiformes, in general, develop viraemia levels above 10^6^ PFU/mL following infection and have a high reservoir competence index. Columbiformes, Pelecaniformes, Psittaciformes, and Galliformes are considered to have a low reservoir competence index. However, there are inter-specific variations in WNV competence, evident in variable levels of viraemia in experimental infections [13], that suggest a differential role of species within the bird community in the local dynamics of WNV. Some authors found little relevance of bird species diversity in the dynamics of WNV and showed that variations in host vector preference is a better driver of virus exposure dynamics [26]. However, exploring the role of the local diversity of competent reservoir birds for WNV may be relevant for understanding patterns of virus exposure at different spatial scales, as observed in mosquito USUV infection probability [27]. A generally underexplored parameter in the spatial dynamics of WNV exposure is variation in the abundance of competent reservoirs (but see [27] for USUV). 

To estimate bird species richness, diversity and abundance, biannual censuses were carried out in the 15 study areas during the mosquito season (spring and autumn). A 1-km-long transect per area was designed to run close to the main water source for mosquito breeding and where birds were captured. The transect was divided into 10 consecutive stretches of 100 m in length. Four bird sighting and listening stations were distributed along each 1-km-long transect with a separation of 300 m. Two dawn and two dusk counts per census were carried out. Two researchers experienced in bird visual and audio identification walked at slow path along each transect carrying binoculars, recording the birds sighted or heard on each 100 m stretch both within a 25 m distance band perpendicular to the transect path (25 m in each side of the transect) and outside that band [28]. The same experienced observers conducted bird counts for 10’ at each station, recording birds within and outside a 25-m radius from the observer. To minimize double counts, we established a temporal separation between stations and transect stretches, and between consecutive stretches, of 5’. Bird richness per study area was estimated as the sum of bird species identified in that area in the different counts and censuses (both within and outside 25 m from the observer). The Shannon diversity index (SDI) was estimated by considering birds counted within 25 m from the observer along the 1-km transects and averaging SDI values from different counts to measure bird diversity per study area. Bird abundance was estimated at the Order taxonomic level to fit the expected differences in reservoir competence for WNV. To do this, we summed the bird species per Order sighted or heard within 25 m from the observer at each of the four stations and calculated the average number of individuals within Order per station as an index of abundance. Final abundance per Order and area was estimated as an average of the values for each count conducted.

#### 2.5.2. Mammal Population Diversity and Abundance

A high number of mammalian species are susceptible to WNV infection, but their role in virus dynamics is limited [29]. However, the local abundance of mammals may be indirectly relevant to the risk of exposure of birds to WNV by driving vector population dynamics [15]. In the study area, the two relevant species in WNV transmission are *Cx. pipiens* and *Cx. theileri* [21]. While *Cx. pipiens* is preferentially ornithophilic, but it may display a wide host range [15], *Cx. theileri*, in contrast, has mammal feeding preferences but can also feed on birds [15]. 

To estimate the frequency of use per area by different mammalian species (see [30]), we placed a camera-trap per study point near the main water source for mosquito breeding attached to wooden sticks at a height of 40 cm above the ground. The cameras (TROPHY CAM HD, Bushnell Outdoor Product, Overland Park, KS, USA) remained active for three fortnight periods along the study (summer and autumn 2018, and spring 2019). To monitor the variation in detectability of animals of different size and weight [31], we placed markers (stones or wooden sticks) at 5-m and 10-m distances from the cameras. The correct functioning of the cameras was monitored by automatically recording images twice a day. The images were classified per animal species. If individuals of different species were observed in an image, this was considered for every species recorded in it. For large mammals, records were truncated to those within 10 m from the camera to control for variations in detectability (e.g., between a large 150 kg red deer stag and a 25 kg roe deer buck). For small mammals, only records made within 5 m from camera-traps were considered. For each mammal species we estimated the frequency of use of the camera space per individual in relation to the time the camera was in operation. Series of images taken consecutively for a species in a time interval of less than 10’ were considered as visits. The duration of each visit (in minutes) was estimated as the time interval between the initial and final images, and this time was multiplied by the maximum number of individuals of the species in the group during the visit. The final sum of visit times for the same camera was used to estimate the species frequency of use in relation to the total time a camera was active in a study area. This index would resemble the time an individual of a species is available to feed mosquitoes in the study area. 

To estimate the abundance and diversity of micromammal species per sampling area, we conducted two (autumn 2018 and spring 2019) trapping events on two consecutive nights per event. Six trapping stations were placed in each area separated by a minimum distance of 200 m. Each trapping station consisted of eight Sherman traps (H.B. Sherman Traps Inc., Tallahassee, FL, USA) placed in two rows of 4 traps each and with a separation between rows and traps of 5 m. The traps remained active from dusk until being checked at dawn on each sampling day. Captured animals were identified to species, and temporarily marked with a 1 cm^2^ shaving on the right thigh to identify recaptures. The number of recaptures was negligible, so we employed the number of captures per trap and night as an index of abundance. 

#### 2.5.3. Mosquito Vector Abundance

The population of mosquitoes per study area was monitored in a fortnight basis between May-December 2018 and April-July 2019. Within a 30m linear distance to the main water source available in a study area, a mosquito trapping station, containing a CDC miniature-type white light trap (John W. Hock Company, Gainesville, FL, USA) and a BG Sentinel trap (BG-Sentinel 2, Biogents, Regensburg, Germany), was set to be active from dusk to dawn for one night. The traps were baited with a CO_2_ slow delivering device provided with 600 g of dry ice and a cold accumulator previously frozen at −80 °C in a 2 L cool container (The Coleman Company Inc., Chicago, IL, USA). Traps were checked at dawn and the captured mosquitoes preserved at ambient temperature in controlled air humidity conditions during transport. Mosquitoes were identified to the species level by morphological traits [15] and preserved frozen at −80 °C. Only data for *Culex* spp. mosquitoes were employed in this study because they are main vectors of *Flavivirus* spp. [3] and because they accounted for 82.8% of the mosquitoes captured in the study sites (Appendix A). The total number of captures per area along the monitoring period were herein employed as vector abundance estimates.

#### 2.5.4. Weather Determinants

Abiotic environmental conditions are one of the most important groups of factors affecting the population dynamics of *Flavivirus* vectors [11]. In addition, the temperature modulates the replication capacity of flaviviruses within vectors and the rate at which they reach levels of viral replication large enough to infect their hosts [32]. At the spatiotemporal scale of our study, these constraints could be reflected in the risk of exposure of birds to flaviviruses determined by conditions that, over a recent time period, have been determinant for the mosquito population, especially as antibodies to these viruses are not long-lived and the half-life of many of the Passeriformes birds studied is not long [14]. Therefore, we collected data from the meteorological stations that were closest to each of the five sampling sites between 2016 and 2018 from the Spanish Meteorological Agency (AEMET) and estimated a series of averaged weather predictors for those years. 

#### 2.5.5. Risk Analysis

We modelled the individual risk of exposure to the virus of birds with a series of predictors potentially modulating spatial variation in risk at the study scale: (i) mosquito host diversity and abundance; (ii) virus reservoir diversity and abundance; (iii) mosquito vector abundance; and (iv) vector and virus abiotic constraints. We selected potential predictors from those described above (Table 1) after a thorough descriptive analysis [33]. All continuous predictors were re-scaled before modelling by subtracting the mean of the predictor to each value and dividing by the standard deviation. After the exploratory analysis, we included the selected predictors and performed all linear model combinations in a mixed approach in which the common collection site of all birds sampled at each of the five study points was considered as a random factor in the modelling process. We selected the models with the best goodness-of-fit to variation in WNV exposure risk by ranking them according to the Akaike information corrected criterion (AICc) using the ‘dredge’ function of the MuMIn R package, and those models with an AICc difference (ΔAICc) below two were included in a model averaging process to obtain the final best-fit model. 

The initial model was run with the original bELISA results without applying any correction for potential cross-reactions with other *Flavivirus* spp. Next, we performed two additional models using response variable data modified according to the minimum (50%) and maximum (95.2%) WNV vs. other *Flavivirus* spp. ratio values estimated from the different studies carried out on wild birds in Spain (Appendix A). The expected prevalence of WNV (at low and high levels of WNV predominance over other flaviviruses) was modified by randomly assigning a continuous value to every individual within each of the 15 surveyed groups (one per survey area) and, thereafter, grouping them in individual random groups (continuous discrete value) in Excel. We then estimated the number of WNV-positive individuals (rounded to unity) that would correspond to each group depending on the low or high proportion of WNV vs. other flaviviruses and assigned as WNV positive the corresponding individuals following the ascending order of the random group value obtained in Excel. For example, if in area A_1_ of S_2_ we identified 4 positive birds out of 34 tested, only two would be WNV positive in a low WNV predominance scenario and 4 would be positive in a high-predominance scenario. The first two birds of the random series in this area were assigned as positive for the model performed assuming the lowest bELISA specificity for WNV, while the first four birds of the random series were considered positive for the model with the highest expected bELISA specificity for WNV. All models were performed according to the abovementioned approach.

## 3. Results

A total of 561 wild birds of more than 40 species belonging to 27 different families, were captured. A major part of the birds (n = 176) belonged to the Paridae family, followed by the families Passeridae (n = 86), Silviidae (n = 44) and Turdidae (n = 41) (Appendix A). The number of individuals captured slightly varied among the three different wildlife-livestock interaction areas allocated to the five study sites, including 174 birds captured in A_1_, 203 in A_2_, and 184 in A_3_ (Table 2). There was also slight variation in the number of captures among study sites, with 98 birds in S_1_, 161 in S_2_, 107 in S_3_, 94 in S_4_, and 101 in S_5_.

Serological testing by bELISA could be carried out on 436 of these birds, yielding 6.9% seropositivity (30 of 436; Clopper-Pearson exact 95% confidence interval (CI): 4.6–9.7%). In the lower WNV vs. other flaviviruses ratio scenario (50%), 15 of the 436 birds would be WNV seropositive (3.4%; CI: 1.9–5.6%). In the scenario of highest WNV predominance (95.2%), 29 birds would be WNV seropositive (29 of 436, 6.7%; CI: 4.5–9.4%). 

The highest number of bELISA positive birds was detected in A_1_ (see Table 2), with 14 of 155 captured birds (9%; CI: 5.0–14.7%) showing the presence of antibodies in comparison to eight of 143 (5.6%; CI: 2.4–10.7%) in A_2_, and eight of 138 (5.8%; CI: 2.5–11.1%) in A_3_. *Flavivirus* seroprevalence was highest in sites S_5_ (11 of 92; 12.0%, CI: 6.1–20.4%) and S_1_ (six of 61; 9.8%, CI: 3.7–20.2%) when compared to sites S_2_ (four of 119; 3.4%, CI: 0.9–8.3%), S_3_ (four of 82; 4.9%, CI: 1.3–12.0%) and S_4_ (five of 82; 6.1%, CI: 2.0–13.7%). Those positive sera that could be analyzed by VNT (n = 5) showed inconclusive results because similar neutralization titers were found for both WNV and USUV. The serum of a blackbird (*Turdus merula*) collected in S_1_ in 2019 displayed a 1:640 VNT titer for USUV and 1:320 for WNV, suggesting a possible USUV infection, but it could not be confirmed as specific for USUV infection because the VNT titer for this virus was not fourfold the obtained for WNV. Specific area and site results for low and high WNV predominance estimates are shown in Table 2.

Four of the 498 animals tested by rRT-PCR were positive for JEV-complex flavivirus RNA (0.8%, CI: 0.2–2.0%; Table 2); two feather pulp samples and two oral swab samples. rRT-PCR positive birds belonged to families Emberizidae (*Emberiza cirlus*, n = 1), Turdidae (*T. merula*, n = 2) and Muscicapidae (*Muscicapa striata*, n = 1) (Table 3). Except for a blackbird captured in the horse farm area (A_1_) in S_3_, the rest were captured in A_2_ areas of sites S_2_, S_3_ and S_5_ (Appendix A). Only one of the four rRT-PCR positive birds, with a Ct value of 15, could be sequenced, indicative of an active infection at the time of capture (September 2018). The sample belonged to a bELISA negative blackbird captured in A_2_ of S_2_ and the sequences (259 and 382 nucleotides long) displayed >99% homology with USUV (GenBank under accession numbers ON758918 and ON758919). The partial sequence ON758919 clustered within the Africa 3 lineage of USUV (Figure 3). Only this bird was positive in the mrRT-PCR with a Ct value of 39.4 for USUV. The other three rRT-PCR positive samples were also negative both in the mrRT-PCR and in the bELISA. 

The analysis of the determinants of the risk of exposure of birds to *Flavivirus* spp. identified a total of eight models with a ΔAICc < 2 that were averaged (Table 4). The averaged model obtained included several predictors related with (1) bird diversity, (2) the gradient of wild bird-horse interaction, (3) the abundance of competent (Passeriformes) and non-competent (Columbiformes) WNV reservoir birds, (4) the abundance of *Culex* spp., and (5) the accumulated rainfall along the year before sampling. However, the only predictor that showed a statistically significant effect on the probability of exposure to *Flavivirus* spp. was the Shannon diversity index for birds (Table 5). The observed effect was a higher risk of exposure in relation to a higher bird diversity estimated at the spatial scale of the sampling area in each study location. The average model obtained for the lower predominance of WNV vs. other flaviviruses included predictors related with (1) bird diversity, (2) the abundance of competent (Passeriformes) WNV reservoir birds, (3) the abundance of *Culex* spp., and (4) the accumulated rainfall along the year before sampling (Table 5). None of the predictors showed a statistically significant influence on the risk of exposure to WNV. The average model for the scenario of high WNV predominance included predictors related with (1) bird diversity, (2) the abundance of Columbiformes, (3) the abundance of *Culex* spp., and (4) accumulated rainfall along the year before sampling. Similar to the general *Flavivirus* spp. model, the Shannon bird diversity index also had a statistically significant effect on WNV exposure risk in this scenario.

## 4. Discussion

This study covers an extensive area in south-central Spain dominated by continental Mediterranean climatic conditions where West Nile fever cases in wildlife and livestock were reported recently [17]. WNV has been detected in mammals, birds, and mosquitoes in Spain, even outside periods of high competent vector activity, so many authors define the virus as endemic in the country [22]. We found *Flavivirus*-infected wild birds in the vicinity of equine farms, confirming their active circulation in the region and the existing risks for new West Nile and Usutu cases or outbreaks. Wild birds living in or in close proximity to horse farms in the region show higher exposure rates to *Flavivirus* spp. and WNV than those living outside the farm environment and not directly linked with farm premises. Our results thus provide novel and useful information for wildlife–livestock interaction scenarios experiencing changes, potentially promoting increasing vector populations in continental Spain. We conclude that continental Mediterranean Spain provides favorable conditions for the circulation of WNV and USUV.

The detection of *Flavivirus* RNA is a difficult task, since viraemia in birds lasts only 1–5 days [34] and viruses can only be detected for a very short period after the end of viraemia in the pulp of growing feather cannons [13]. Consequently, and as expected, only four of the 503 birds analyzed in our study were PCR positive. The employed duplex rRT-PCR protocol has a high detection sensitivity of below 50 RNA copies for both Japanese encephalitis and Ntaya *Flavivirus* spp. serogroups [24], so our findings indicate a low incidence of recent infections (positivity prevalence range: 1.7–4.5%). This low incidence prevented confirming PCR positive samples by sequencing. Only a sample with a very low Ct could be sequenced and confirmed as USUV. This finding constitutes the first detection of USUV RNA in a naturally infected blackbird in Spain and one of the few recent records of Africa 3 lineage in the country [35,36]. Previously, USUV RNA was sequenced in Spanish song thrushes (*Turdus philomelos*) in 2012 [20]. Recently, Bravo-Barriga et al. [37] and Marzal et al. [38] confirmed USUV antibodies in exotic (*Euplectes afer*) and native birds (*Gyps fulvus*, *Bubo bubo,* and *Otis tarda*), and Guerrero-Carvajal et al. [5] also confirmed exposure in horses from western Spain, but they did not report USUV PCR positive results. We were unable to confirm the JEV group flaviviruses that were infecting the other three PCR positive birds, but all three were bELISA negative. Considering the time required for infected birds to seroconvert, these were thus very recent infections confirming active circulation of JEV sero-complex flaviviruses in the area and time of study. 

Available volume of blood that can be extracted in birds is limited, especially from small passerine birds, limiting the volume of plasma to confirm seropositive samples. Cross-reactions to different flaviviruses of the JEV group occur in commercial ELISA tests [23], limiting their specificity. However, all recent serological studies performed on different animal species in Spain systematically show that a high proportion of the positive samples (tested also with the INGEZIM West Nile COMPAC bELISA) have specific antibodies (measured by VNT) against WNV (range: 19–70%) in contrast to a lower proportion of USUV exposure (range: 1–26%) [5,37,38]. The proportion of VNT-confirmed WNV vs. other flaviviruses did indeed show that WNV is the predominant *Flavivirus* spp. in wild birds in Spain, with four extensive studies showing 94–100% confirmed WNV predominance and two more limited studies (in species number/distribution range or sample size) showing similar circulation rates of both WNV and USUV. Further support to the predominance of WNV with respect other flaviviruses, unpublished bELISA and VNT serological results from non-vaccinated horses in our study farms, also confirms the predominance of antibodies to WNV over USUV and the active but silent circulation of WNV. Therefore, despite the limited number of VNT tested samples, the authors highly suspect that the highest proportion of bELISA seropositive birds would have been exposed to WNV. The average exposure rate to WNV in continental Spain (3.4% to 6.7%) is in line with that reported in other European countries such as Italy (4.3%), France (4.8%), Czech Republic (5.9%) and Serbia (7.6%) [37]. Slightly higher prevalence is reported for wild birds in western (12.7–18.2%) [37,38] and southern Spain (15.2%) [39] where cases in horses, humans, or birds have been reported with a higher frequency than in our area [17]. Other studies report local seroprevalence rates above 50% in waterfowl from a marsh environment with high mosquito densities [40] that are not comparable to the low densities obtained in continental Spain [21] (vide infra). The passive method of capture with mist nets is efficient and safe for the studied birds. However, it prevents from capturing waterfowl, birds of prey and other large wild birds involved in WNV ecology. Nonetheless, waterfowl is not abundant outside wetlands in continental Spain and raptors are much less abundant than passerine birds, so our survey constitutes a good representation of wild birds in and around horse farms. An interesting finding was that only one of 30 (3.3%) analyzed corvids was bELISA positive; it was indeed one of the lowest seroprevalence rates of the studied bird families (Table 3). The high susceptibility of corvids to WNV could result in a reduced survival of infected individuals, and thus in the low seroprevalence found [13]. The structure of the community of birds may have local influences over the transmission dynamics of flaviviruses that need to be explored. 

Bird diversity at local spatial scales was not a relevant risk factor for exposure of birds to WNV in the US [26] whereas Ferraguti et al. [41] found a positive association instead that was in line with our modelling findings. Diversity was highest in S_1_ and S_5_ where highest overall exposure rates were observed, thus clearly explaining the observed association. None of the previously mentioned studies was able to address whether changes in avian community assemblages could be responsible for the contrasting findings. In our study settings, Passeriformes were predominant over the rest of orders (Appendix A), and we found no relevant contrast among study sites and areas in the structure of the avian community suggestive of a specific role of a particular family of Passeriformes. However, Sturnidae and Passeridae were, in general terms, the predominant bird families across our study area, and they were together the most abundant birds in the areas with the highest seroprevalence. *Passer domesticus* and *Sturnus unicolor* were the predominant species within their respective families, and both were highly abundant in or close to the farm environment (A_1_ and A_2_ areas). The farm environment that attracts different Passeriformes may promote the aggregation of individuals within a favorable environment for the flourishment of *Culex* spp. vectors, and thus enhance the transmission of WNV and other viruses. Aggregation rather than abundance could be the relevant trait favoring transmission of flaviviruses in farm scenarios. That would also explain why the estimated order-specific bird abundance indices showed a low relevant role in virus exposure risk. In addition, farms can be expected to provide a good roosting refuge for birds during the hours when mosquitoes are most active. Off-farm, roosting aggregations of birds should be lower, and therefore we observed a slightly lower risk of exposure than in the farm environment. 

The abundance of vectors is a relevant parameter for pathogen transmission [42] that we did not clearly evidence in our settings. The low and relatively similar abundance observed in study settings could perhaps render its low relevance in continental Mediterranean climates. This would also relate to the positive, albeit not strong, influence of previous year accumulated rainfall on exposure risk. However, a relevant aspect to consider in the interpretation of the associations of predictors with exposure risk is the persistence of detectable levels of specific antibodies in birds, since in some cases it can exceed three years [14], i.e., almost lifelong immunity for some short-lived species. Thus, birds that were exposed to the bite of infected mosquitoes in years prior to sampling could still be seropositive when captured for this study. We measured mosquito abundance across 2018 and early 2019, so we might have underestimated the relevance of *Culex* spp. abundance on WNV exposure risk with our dataset if that exceeded the usual in 2017 when a peak of virus transmission could have occurred. Accumulated rainfall over the 2017–2018 agro-hydrological year exceeded the normal rainfall accumulation values along the climate series 1981–2010. In contrast, 2018–2019 was one of the driest years in the Spanish series (www.aemet.es, accessed on 29 August 2022), which might have accounted for the low *Culex* spp. abundance estimated in our fortnight survey. Other evidence supporting a peak of *Flavivirus* transmission in 2017 includes the observation of a peak in the incidence of exposure of wild ungulates in the study areas in that year (the authors, unpublished data). 

## 5. Conclusions

We can conclude that in areas where interactions between wild birds and livestock occur frequently, the likelihood of exposure to flaviviruses seems to be higher than in areas where the interactions are less common or almost inexistent. This results in an elevated risk of infection for horses and humans on farm environments, even in continental Mediterranean areas where climatic conditions suffer extreme changes both seasonally and interannually that limit vector abundance. Birds roosting in farm environments should be considered as an important risk for WNV maintenance and transmission to livestock and humans. Diverting the attention of habituated birds away from the farm environment, e.g., by relocating grain stores from the stables or favoring roosting sites for birds outside the farm buildings, could be measures to mitigate the risk of infection. The seemingly unstoppable expansion of WNV cases in the Iberian Peninsula suggests that such measures should be adopted preventively throughout south-central Spain. Perhaps the early adoption of measures, while not preventing all cases, will serve to reduce the incidence and impact of WNV.

## Figures and Tables

**Figure 1 pathogens-12-00083-f001:**
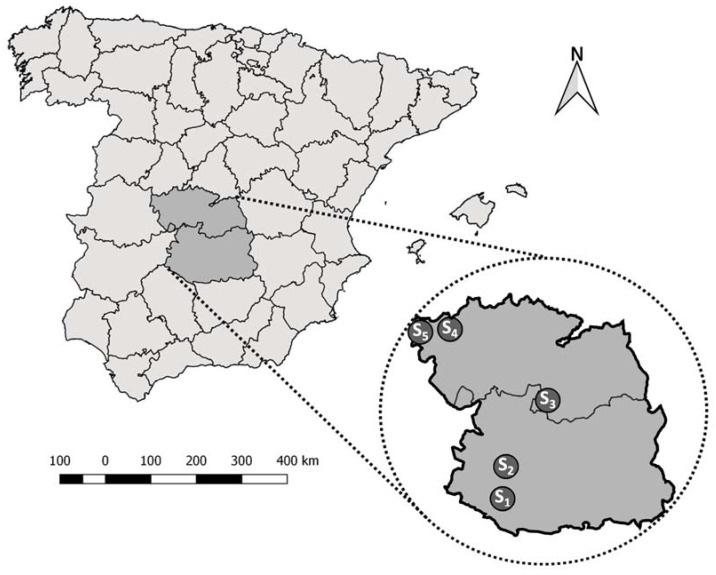
Location of the five selected study sites (S_1_ to S_5_) within Toledo and Ciudad Real provinces in peninsular Spain.

**Figure 2 pathogens-12-00083-f002:**
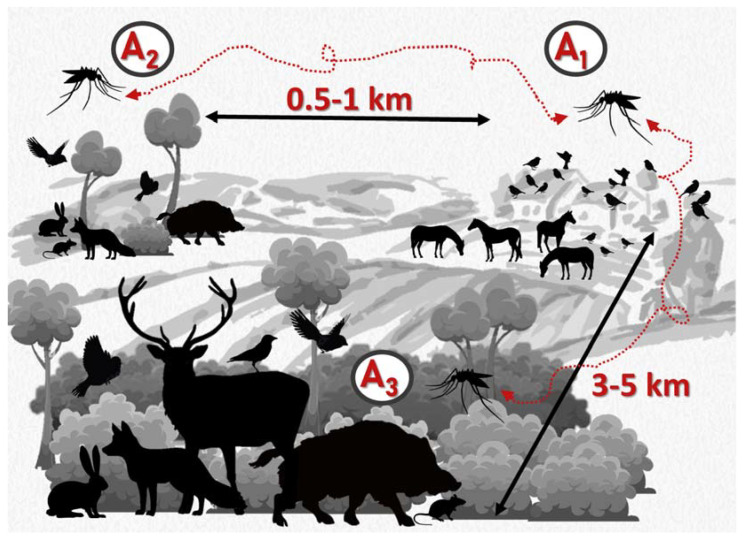
Schematic representation of the different epidemiological scenarios (A_1_, A_2_ and A_3_) selected for the study based on a varying gradient of wildlife-livestock interaction.

**Figure 3 pathogens-12-00083-f003:**
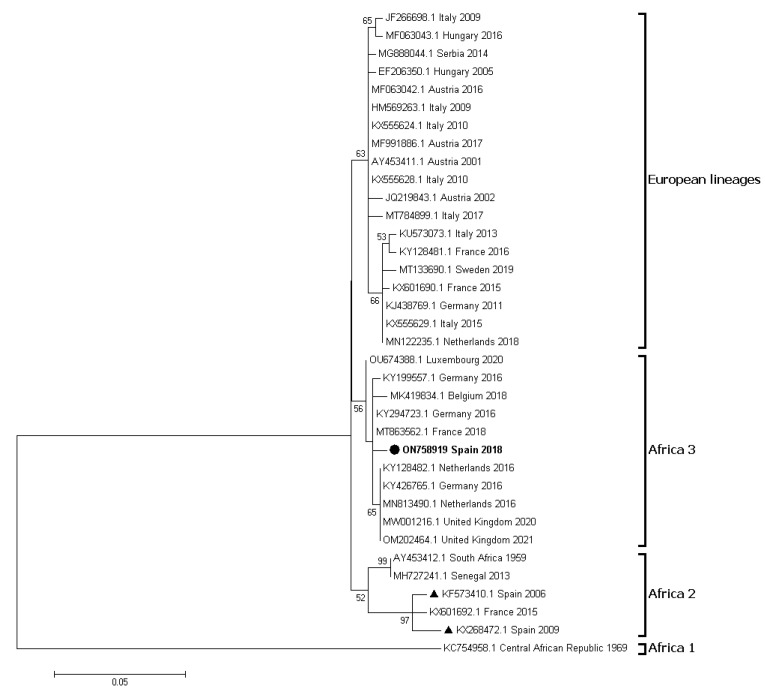
Phylogenetic analysis of Usutu virus (USUV) strains. Phylogenetic analysis based on 36 partial nucleotide sequences (382bp, ON758919) of USUV. USUV sequences are identified by GenBank accession number, country and year of isolation. Sequence emphasized in bold and with a circle was generated during this study. Other Spanish strains are marked with a triangle. Percentages of successful bootstrap replicates over 50% are indicated at tree nodes.

**Table 1 pathogens-12-00083-t001:** Set of predictors gathered for statistical analyses and range values. The predictors included in multiple risk factor modelling are highlighted in bold type.

Factor	Predictor	Description	Type & Values
Spatial	**Site**	**Survey location**	**Categorical (1–5)**
	**igrad**	**Interaction gradient**	**Categorical (1–3)**
Bird host population	ab.tot	Bird average abundance index	Numerical (4.2–170)
	**ab.pass**	**Abundance of passerine bird index**	**Numerical (3.6–167)**
	**ab.col**	**Abundance of columbiform bird index**	**Numerical (0.08–3.02)**
	ab.bucer	Abundance of bucerotiform bird index	Numerical (0–0.65)
	ab.picif	Abundance of piciform bird index	Numerical (0–0.27)
	ab.accip	Abundance of accipitriform bird index	Numerical (0–0.31)
	ab.corac	Abundance of coraciiform bird index	Numerical (0–5.25)
	ab.sulif	Abundance of suliform bird index	Numerical (0–0.06)
	ab.cicon	Abundance of ciconiform bird index	Numerical (0–0.15)
	rich	Bird richness	Numerical (16–36)
	**ish**	**Shannon diversity index**	**Numerical (0.18–2.77)**
	ismp	Simpson diversity index	Numerical (0.07–2.01)
Ungulate population	ab.ung	Ungulate abundance index	Numerical (0.00–0.02)
Vector population	**cx**	** *Culex * ** **spp. abundance index**	**Numerical (2–103)**
Weather	ar	Annual accumulated rainfall	Numerical (361.1–739.4 mm)
	swr	Rainfall accumulated over Dec–May	Numerical (185.0– 553.6 mm)
	sr	Rainfall accumulated in spring (March–May)	Numerical (127.0– 355.6 mm)
	smr	Summer (July–Sept) accumulated rainfall	Numerical (16.7– 166.5 mm)
	wt	Average winter (Dec–Feb) temperature	Numerical (9.8–15.9 °C)
	st	Average spring (March–May) temperature	Numerical (9.8–15.9 °C)
	smt	Average summer (July–Sept) temperature	Numerical (18.6–27.9 °C)
	**ar1**	**Annual cumulative rainfall of year t-1**	**Numerical (362.1–856.5 mm)**
	swr1	Rainfall accumulated over Dec-May year t-1	Numerical (208.4–502.4 mm)
	sr1	Rainfall accumulated in spring year t-1	Numerical (100.2–249.1 mm)
	smr1	Summer accumulated rainfall year t-1	Numerical (2.0–60.4 mm)
	wt1	Average winter temperature year t-1	Numerical (3.4–8.5 °C)
	st1	Average spring temperature year t-1	Numerical (9.0–17.1 °C)
	smt1	Average summer temperature year t-1	Numerical (18.7–27.9 °C)
	ar2	Annual cumulative rainfall of year t-2	Numerical (353.4–497.2 mm)
	swr2	Rainfall accumulated over Dec-May year t-2	Numerical (175.8–372.1 mm)
	sr2	Rainfall accumulated in spring year t-2	Numerical (69.0–249.1 mm)
	smr2	Summer accumulated rainfall year t-2	Numerical (2.0–90.6 mm)
	wt2	Average winter temperature year t-2	Numerical (5.2–9.3 °C)
	st2	Average spring temperature year t-2	Numerical (11.2–15.2 °C)
	smt2	Average summer temperature year t-2	Numerical (19.7–28.1 °C)

**Table 2 pathogens-12-00083-t002:** Number of birds captured and analyzed by blocking ELISA and rRT-PCR per area (A_1_–A_3_) and site (S_1_–S_5_). Number of birds tested and positive to bELISA and rRT-PCR to estimate antibody and infection prevalence are shown. The corrected WNV antibody prevalence under low (50%) and high (95.2%) predominance over other *Flavivirus* spp. scenarios is shown as bELISA_low_ and bELISA_high_, respectively.

Area	Site	No. Captures	bELISA No. Positive/No. Tested(Prevalence)	bELISA_low_ No. Positive/No. Tested(Prevalence)	bELISA_high_ No. Positive/No. Tested(Prevalence)	rRT-PCR No. Positive/No. Tested (Prevalence)
A_1_	S_1_	36	4/34 (11.8%)	2/34 (5.9%)	4/34 (11.8%)	0/32 (0.0%)
	S_2_	56	2/48 (4.2%)	1/48 (2.1%)	2/48 (4.2%)	0/43 (0.0%)
	S_3_	24	3/18 (16.7%)	1/18 (5.6%)	3/18 (16.7%)	1/22 (4.5%)
	S_4_	24	3/23 (13.0%)	2/23 (8.7%)	3/23 (13.0%)	0/22 (0.0%)
	S_5_	34	2/32 (6.3%)	1/32 (3.1%)	1/32 (3.1%)	0/32 (0.0%)
Subtotal A_1_		174	14/155 (9.0%)	7/155 (4.5%)	13/155 (8.4%)	1/151 (0.7%)
A_2_	S_1_	28	0/7 (0.0%)	0/7 (0.0%)	0/7 (0.0%)	0/28 (0.0%)
	S_2_	67	2/44 (4.5%)	1/44 (2.3%)	2/44 (4.5%)	1/58 (1.7%)
	S_3_	45	1/32 (3.1%)	1/32 (3.1%)	1/32 (3.1%)	1/37 (2.7%)
	S_4_	25	1/25 (4.0%)	0/25 (0.0%)	1/25 (4.0%)	0/24 (0.0%)
	S_5_	38	4/35 (11.4%)	2/35 (5.7%)	4/35 (11.4%)	1/30 (3.3%)
Subtotal A_2_		203	8/143 (5.6%)	4/143 (2.8%)	8/143 (5.6%)	3/177 (1.7%)
A_3_	S_1_	34	2/20 (10.0%)	1/20 (5.0%)	2/20 (10.0%)	0/33 (0.0%)
	S_2_	38	0/27 (0.0%)	0/27 (0.0%)	0/27 (0.0%)	0/34 (0.0%)
	S_3_	38	0/32 (0.0%)	0/32 (0.0%)	0/32 (0.0%)	0/32 (0.0%)
	S_4_	45	1/34 (2.9%)	1/34 (2.9%)	1/34 (2.9%)	0/45 (0.0%)
	S_5_	29	5/25 (20.0%)	2/25 (8.0%)	5/25 (20.0%)	0/26 (0.0%)
Subtotal A_3_		184	8/138 (5.8%)	4/138 (2.9%)	8/138 (5.8%)	0/170 (0.0%)
All areas	S_1_	98	6/61 (9.8%)	3/61 (4.9%)	6/61 (9.8%)	0/93 (0.0%)
	S_2_	161	4/119 (3.4%)	2/119 (1.7%)	4/119 (3.4%)	1/135 (0.7%)
	S_3_	107	4/82 (4.9%)	2/82 (2.4%)	4/82 (4.9%)	2/91 (2.2%)
	S_4_	94	5/82 (6.1%)	3/82 (3.7%)	5/82 (6.1%)	0/91 (0.0%)
	S_5_	101	11/92 (12.0%)	5/92 (5.4%)	10/92 (10.9%)	1/88 (1.1%)
**Total**		**561**	**30/436 (6.9%)**	**15/436 (3.4%)**	**29/436 (6.7%)**	**4/498 (0.8%)**

**Table 3 pathogens-12-00083-t003:** Summary of the results of serological and molecular (feather cannon pulp, oral swab or cloacal swab) analyses of bELISA seropositive captured wild birds across survey sites and interaction scenarios. Results are presented at the bird Family taxonomic level.

Bird Family	bELISA No. Positive /No. Tested (Seroprevalence)	rRT-PCR No. Positive /No. Tested (Prevalence)
*Corvidae*	1/29 (3.4%)	0/28 (0.0%)
*Emberezidae*	1/13 (7.7%)	1/19 (5.3%)
*Fringilidae*	3/27 (11.1%)	0/25 (0.0%)
*Muscicapidae*	0/15 (0.0%)	1/14 (7.1%)
*Paridae*	14/124 (11.3%)	0/167 (0.0%)
*Passeridae*	5/82 (6.1%)	0/77 (0.0%)
*Turdidae*	6/40 (15.0%)	2/32 (6.3%)
** *TOTAL* **	**30/330 (9.1%)**	**4/362 (1.1%)**

**Table 4 pathogens-12-00083-t004:** Set of models with ΔAICc < 2 selected for model averaging of the general bELISA (*Flavivirus* spp.) model and the WNV low (WNV_low_) and high (WNV_high_) predominance over other flaviviruses.

Model Set	Model Reference	ΔAICc/ Weight	Predictors					
			igrad	ab.col	ab.pass	cx	ish	ar1
*Flavivirus* spp.	model 1	0.00/0.088	ns	ns	ns	ns	0.6765	ns
	model 2	0.07/0.085	ns	ns	ns	ns	0.5703	0.2433
	model 3	0.53/0.068	ns	ns	ns	0.2211	0.5498	ns
	model 4	1.36/0.045	+	−0.5336	ns	ns	0.7767	0.5429
	model 5	1.39/0.044	ns	ns	−0.1983	ns	0.5159	0.3965
	model 6	1.41/0.043	ns	−0.1938	ns	ns	0.6571	0.3313
	model 7	1.82/0.036	ns	ns	0.07899	ns	0.6691	ns
	model 8	1.98/0.033	ns	ns	ns	0.09002	0.5452	0.1825
WNV_low_	model 1	0.00/0.122	ns	ns	ns	ns	0.5262	ns
	model 2	0.84/0.080	ns	ns	ns	ns	ns	ns
	model 3	1.67/0.053	ns	ns	ns	ns	ns	0.2632
	model 4	1.80/0.050	ns	ns	ns	ns	0.4735	0.1223
	model 5	1.93/0.046	ns	ns	ns	0.2370	ns	ns
	model 6	1.95/0.046	ns	ns	0.06779	ns	0.5170	ns
	model 7	1.97/0.045	ns	ns	ns	0.0664	0.4904	ns
WNV_high_	model 1	0.00/0.145	ns	ns	ns	ns	0.6818	ns
	model 2	1.06/0.085	ns	ns	ns	0.1838	0.5746	ns
	model 3	1.52/0.068	ns	ns	ns	ns	0.6282	0.1291
	model 4	1.79/0.059	ns	−0.1041	ns	ns	0.7494	ns

* ns: predictors not included in models with ΔAICc < 2.

**Table 5 pathogens-12-00083-t005:** Output of average selected models including predictors (Abbreviations shown in Table 1), estimates and their associated standard error (SE), the statistic (z) and the *p*-value. The model set testing bELISA positivity risk is shown as *Flavivirus* spp. model, whereas the models for the low and high predominance of WNV over other flaviviruses are named WNV_low_ and WNV_high_, respectively.

Model Set	Predictor	Estimate	SE	z	*p*
*Flavivirus* spp.	Intercept	−2.77232	0.27952	9.899	***
	ish	0.61848	0.29538	2.089	*
	ar1	0.18747	0.24631	0.760	0.447
	cx	0.04062	0.12610	0.322	0.747
	igrad				0.147
	A_1_	Ref.			
	A_2_	−0.11159	0.37507	0.297	0.766
	A_3_	−0.07787	0.29499	0.264	0.792
	ab.col	−0.07307	0.20228	0.361	0.718
	ab.pass	−0.01337	0.10871	0.123	0.902
WNV_low_	Intercept	−3.40845	0.28796	11.804	***
	ish	0.30226	0.36776	0.821	0.412
	ar1	0.04525	0.14548	0.311	0.756
	cx	0.03166	0.13447	0.235	0.814
	ab.pass	0.00703	0.07749	0.090	0.928
WNV_high_	Intercept	−2.81347	0.22789	13.311	***
	ish	0.65721	0.28526	2.298	*
	ar1	0.02446	0.09261	0.264	0.792
	cx	0.04395	0.11972	0.366	0.714
	ab.col	−0.01728	0.09418	0.183	0.855

* *p* < 0.05; ** *p* < 0.01; *** *p* < 0.001.

## Data Availability

Data are available upon reasonable request to the authors.

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
