# Peer review of "Risk Factors for Exposure of Wild Birds to West Nile Virus in A Gradient of Wildlife-Livestock Interaction"

_pathogens, 2023, doi:10.3390/pathogens12010083_

Round 1

Author Response

  • Overall, this is an excellent study documenting the increased seroprevalence to flavivirus infection (e.g., West Nile virus) in wild bird populations inhabiting the farm environment, as opposed to wild bird populations living in areas outside the farm environment. Authors suggest that the risk of flavivirus infection to both humans and livestock within the farm environment may also be higher than it is outside the farm environment. As authors point out, a dominant mosquito species in this region is Culex pipiens, as reported in Ref. 21 (Duran-Martinez 2012 Ph.D. dissertation). Importantly, Culex pipiens can serve as both the enzootic vector among birds and a bridge vector to livestock and humans. For example, in a biome like that described in this study (i.e., Portugal), Osario et al. 2012 found that among 78 wild-caught blood-fed Culex pipiens, most (70%) had fed on birds, but a substantial proportion (23%) also fed on humans.

Authors’ response (AR): Thank you for your positive comments. We have considered and included your suggestions in the revised version of the manuscript.

  • In general, this is a well-conceived, appropriately analyzed, and well-written study. The conclusions are clearly presented and discussed.
  • The only strong recommendation that I offer would be to present results on the entomological component of this system. For example, what was the average mosquito abundance and species composition at A1, A2 and A3? This could be presented as a simple sentence in the RESULTS section, presented as a small table, or even better, the full data could be included as Supplementary Information. With respect to the entomological component, the authors remarks regarding the large difference in accumulated rainfall between 2017-2018 versus 2018-209 (see DISCUSSION; lines 535-542) was very helpful in understanding why the predictor ‘cx’ was not significant in any of their statistical models (Table 5).

AR: Thank you for the suggestion. We understand the relevance of seeing this information in the manuscript to understand the model results. We had not included it previously because the result of mosquito catches at the nine (twenty-seven) study sites are the subject of another paper that has recently been submitted for peer-reviewing. We have now included the detailed information on mosquito captures for the five (fifteen) study sites as a supplementary table.

  • Also, the authors did not say anything about the presence of domestic fowl (e.g., chickens, ducks, quail, etc.) as being part of the farm environment. This is important because these bird species are relatively large compared to songbirds (i.e., greater surface area and CO2 emission [=mosquito attractant]), may or may not be very abundant, and are generally regarded as poor reservoirs of West Nile virus because they produce such low viremia. Thus, domestic poultry on a farm may serve as “dilution hosts” that divert Culex vectors away from more flavivirus- competent Passiformes bird species inhabiting the farm environment. This may be a minor point but at the very least, authors should briefly mention the relative abundance of poultry on the nine farms that were included in this study.

AR: Thanks for the suggestion. We interviewed farm owners/managers/vets before starting our survey to obtain information on the presence of other domestic animals in the farm, among other information. Only in the farm environment at site S1 did they keep a very small number of hens (n=8) that we sampled for blood and tested by WNV ELISA with negative results (not included in the study). Given that the hens were not in the same enclosure as the horses but in another closed building, to their low number with respect to the wild birds present on the farm and to the fact that they were not close to the routes taken for the on-farm bird census, they were not considered in the study. In none of the areas of the rest of the sampling sites was the presence of domestic birds or other domestic species recorded, except for dogs.

  • As authors are probably aware, mosquito control personnel in California routinely set out immunologically naïve ‘sentinel’ chickens and quail, and the proportion of sero-conversions that result are used to assess local levels of flavivirus activity.

AR: Yes, thank you. In fact, domestic birds are subject to surveillance in Spain as well. In relation to your comment, we have recently conducted a study that is in the process of peer review on the usefulness of wild ungulates to monitor the spatiotemporal dynamics of transmission of West Nile virus and other flaviviruses in Spain. The results indicate that this information, which can be easily obtained from the active wildlife health surveillance programs implemented in the country, would allow us to predict the risk of outbreaks in advance. Combining these results with surveillance in domestic birds and even in domestic ungulates in risk environments (equine or poultry farms and urban environments), would generate a very useful tool for prevention and informed decision-making for future actions.

Reviewer 2 Report

Cases of human West Nile neuroinvasive disease in Europe and especially in Spain are increasing and understanding risk factors accounting for the virus amplification and spread are crucial to maintaining appropriate vector control activities among other preventive measures to protect animal and human health. The article “Risk factors for exposure of wild birds to West Nile virus in a gradient of wildlife-livestock interaction” falls in this context. It shows that birds living in equine farms are more exposed to flavivirus infections than outside and that passerine aggregations in and around the farm are a risk factor for higher exposure to WNV. The article reads very well and the conclusions are fairly supported by the results. Only MINOR corrections are suggested:

Line 99 to 101: Can the authors explain why are these farms considered representative of the continental Mediterranean ecosystems?

Line 190: Why the author chose to not conduct PCRs on mosquito pools? Can’t we hypothetically suppose that the rate and level of infection of the local WNV mosquito vector population are what influences the prevalence of WNV in the bird population in the 3 areas?

Lines 191-192: Could the authors explain why did they use the pulp of growing feather cannons and oral and cloacal swabs for the RT-PCR instead of blood? Lines 454-454: and the sensitivity of this test compared to directly detecting Flaviviruses in the blood?

Line 302: “Only data for Culex spp. mosquitoes were employed in this study” Could the authors add a sentence to explain the rationale behind this choice?

Lines 398-399: Could the authors indicate the lineage classification of the sequenced USUV strain? This is really important to track circulating strains and predict the severity of infections both in birds and humans. Data can be added to the discussion starting from line 458

Table 3: Could the authors precise the type of samples used for the PCR in the legend the reader doesn’t understand that the PCR was run on sera like the ELISA?

Line 483-485: Could the authors use this sentence: Despite the limited number of VNT tested samples, the authors highly suspect that the highest proportion of ELISA seropositive birds would have been exposed to WNV.

Author Response

  • Cases of human West Nile neuroinvasive disease in Europe and especially in Spain are increasing and understanding risk factors accounting for the virus amplification and spread are crucial to maintaining appropriate vector control activities among other preventive measures to protect animal and human health. The article “Risk factors for exposure of wild birds to West Nile virus in a gradient of wildlife-livestock interaction” falls in this context. It shows that birds living in equine farms are more exposed to flavivirus infections than outside and that passerine aggregations in and around the farm are a risk factor for higher exposure to WNV. The article reads very well and the conclusions are fairly supported by the results. Only MINOR corrections are suggested:
  • Line 99 to 101: Can the authors explain why are these farms considered representative of the continental Mediterranean ecosystems?

Authors’ response (AR): It is a good question, and the answer is that the study farms are not spatially representative of the whole Spanish Mediterranean continental territory that includes two Plateaus, north and south, both together with an extension of 400,000 of the 505,944 Km2 that the peninsular part of Spain has. What we wanted to express in the manuscript is that the study areas are representative of Mediterranean continental environments in which the presence of West Nile virus has been reported. We have corrected the text to reflect this aspect more clearly.

  • Line 190: Why the author chose to not conduct PCRs on mosquito pools? Can’t we hypothetically suppose that the rate and level of infection of the local WNV mosquito vector population are what influences the prevalence of WNV in the bird population in the 3 areas? 

AR: Thanks for the question. Our plans were indeed to test mosquito pools to estimate flavivirus transmission risk, but we discarded that idea after the low antibody and virus prevalence found in birds and the low number of mosquitoes captured in some of the areas. We considered that we would not be able to detect the presence (and accurately estimate prevalence) of the virus in Culex spp. mosquitoes in our settings given that studies carried out in European regions with higher incidence of West Nile fever resulted in very low prevalence infection rates in mosquitoes (e.g., Vázquez et al., 2011; https://doi.org/10.4269/ajtmh.2011.11-0042; Engler et al., 2013; https://doi.org/10.3390/ijerph10104869).

  • Lines 191-192: Could the authors explain why did they use the pulp of growing feather cannons and oral and cloacal swabs for the RT-PCR instead of blood?

AR: We have included a reference supporting the election of sample type in the text and better clarified this aspect in the discussion. Recent evidence from experimental infection studies indicates that the West Nile viral genome persists for a longer detectable time in the pulp of growing feather cannons than in peripheral blood and for a similar time in secretions (see Pérez-Ramírez et al., 2014; https://doi.org/10.3390/v6020752 and references therein).

  • Lines 454-454: and the sensitivity of this test compared to directly detecting Flaviviruses in the blood? 

AR: This has been better clarified in the discussion. The duplex real-time RT-PCR can detect below 50 RNA copies for both JE and Ntaya serogroups in different samples (Elizalde et al., 2020; https://doi.org/10.3389/fvets.2020.00203). No analysis has been performed to compare the sensitivity of this method as a function of sample type, but the use of other highly sensitive PCR protocols has demonstrated that viral RNA can be detected for a longer period in feather cannons than in blood (Pérez-Ramírez et al., 2014; https://doi.org/10.3390/v6020752, and references therein). In our opinion, it would not be expected that PCR sensitivity would vary according to sample type but that any differences in testing different samples would be a function of the duration of virus genome undegraded in the sample.

  • Line 302: “Only data for Culex spp. mosquitoes were employed in this study” Could the authors add a sentence to explain the rationale behind this choice?

AR: Thank you for the suggestion. We have included a sentence in the manuscript indicating that the main vectors of flaviviruses in our study areas are species of the genus Culex, not only because they are the most competent vectors among the group of mosquitoes that could transmit these flaviviruses, but also because they represent more than 80% of the mosquito captures in the study sites.

  • Lines 398-399: Could the authors indicate the lineage classification of the sequenced USUV strain? This is really important to track circulating strains and predict the severity of infections both in birds and humans. Data can be added to the discussion starting from line 458.

AR: Thank you for such a relevant suggestion. Honestly, we did not delve into the classification of this sequence because the main objective of the work is focused on understanding the determinants of WNV in wildlife-to-anthropized transition environments, but we agree with the reviewer that this information is relevant and provides a finding to better understand the dynamics of USUV in the Palearctic. Classification could not be performed with the short 259bp sequence (GenBank accession number ON758918), but the longer sequence (382bp, ON758919) obtained was aligned using Clustal W with 35 whole genome sequences of USUV registered in GenBank. The results of the phylogenetic tree construction indicated that this sequence clusters within the Africa 3 lineage, widely described in Western Europe (new reference included), but never before in Spain. This information has been included in the revised version of the manuscript and we have also included a new figure (Figure 3) showing the phylogenetic tree of the classification analysis. We have updated the discussion according to this new information, but without going into excessive length because the focus of the study is mainly on WNV.

  • Table 3: Could the authors precise the type of samples used for the PCR in the legend the reader doesn’t understand that the PCR was run on sera like the ELISA? 

AR: Thank you for your suggestion. We have included sample type subject to PCR in table heading.   

  • Line 483-485: Could the authors use this sentence: Despite the limited number of VNT tested samples, the authors highly suspect that the highest proportion of ELISA seropositive birds would have been exposed to WNV.

AR: Thank you for your kind suggestion. This is now included in the manuscript.